# A Single Dose of the Deactivated Rabies-Virus Vectored COVID-19 Vaccine, CORAVAX, Is Highly Efficacious and Alleviates Lung Inflammation in the Hamster Model

**DOI:** 10.3390/v14061126

**Published:** 2022-05-24

**Authors:** Drishya Kurup, Christoph Wirblich, Leila Zabihi Diba, Rachael Lambert, Megan Watson, Noor Shaikh, Holly Ramage, Charalambos Solomides, Matthias J. Schnell

**Affiliations:** 1Department of Microbiology and Immunology, Sidney Kimmel Medical College at Thomas Jefferson University, Philadelphia, PA 19107, USA; drishya.kurup@jefferson.edu (D.K.); christoph.wirblich@jefferson.edu (C.W.); leila.zabihidiba@jefferson.edu (L.Z.D.); rachael.lambert@jefferson.edu (R.L.); megan.watson@jefferson.edu (M.W.); noor.shaik@jefferson.edu (N.S.); holly.ramage@jefferson.edu (H.R.); 2Department of Pathology, Thomas Jefferson University Hospital, Philadelphia, PA 19107, USA; charalambos.solomides@jefferson.edu; 3Jefferson Vaccine Center, Sidney Kimmel Medical College, Thomas Jefferson University, Philadelphia, PA 19107, USA

**Keywords:** SARS-CoV-2, vaccine, efficacy, single dose, lung inflammation, protection, cytokine storm, weight loss

## Abstract

Without sufficient herd immunity through either vaccination or natural infection, the coronavirus disease 2019 pandemic is unlikely to be controlled. Waning immunity with the currently approved vaccines suggests the need to evaluate vaccines causing the induction of long-term responses. Here, we report the immunogenicity and efficacy of our adjuvanted single-dose Rabies-vectored SARS-CoV-2 S1 vaccine, CORAVAX, in hamsters. CORAVAX induces high SARS-CoV-2 S1-specific and virus-neutralizing antibodies (VNAs) that prevent weight loss, viral loads, disease, lung inflammation, and the cytokine storm in hamsters. We also observed high Rabies VNA titers. In summary, CORAVAX is a promising dual-antigen vaccine candidate for clinical evaluation against SARS-CoV-2 and Rabies virus.

## 1. Introduction

Severe acute respiratory syndrome coronavirus type 2 (SARS-CoV-2) was first reported in Wuhan, China, in December 2019 and accounts for over 264 million infections and 5.22 million deaths at the time of this report [1]. SARS-CoV-2 infection in humans causes coronavirus-19 disease (COVID-19), resulting in mild or severe clinical manifestations that mainly affect the respiratory system.

SARS-CoV-2 is an enveloped virus containing the spike (S) glycoprotein on its surface, which is required for viral entry into cells [2]. The S protein is the main target for current vaccine development because antibodies directed against this protein can neutralize the virus. Fast-tracked research efforts by academic institutions and biotech companies led to the development and approval of several SARS-CoV-2 S-protein-based vaccines that induce neutralizing antibodies [2]. Recent studies suggest that neutralizing antibody levels are highly predictive of immune protection from symptomatic SARS-CoV-2 infection [3]. Whereas the previous SARS-CoV was controlled using social isolation and screening, control of the spread of SARS-CoV-2 has proven very challenging. Fast-tracked vaccine development and approval led to vaccine administration by December 2020 and into early January 2021 in several countries. Despite these efforts, controlling the pandemic has been difficult due to several factors: (1) high transmissibility, (2) transmission by asymptomatic individuals, (3) unequal distribution of vaccines worldwide, (4) emergence of variants with higher transmissibility, and (5) vaccine hesitancy.

The first mRNA-based COVID-19 vaccines were approved around December 2020 (BNT162b: Pfizer-BioNTech, USA-Germany; mRNA-1273: Moderna, USA). These vaccines use a lipid-based delivery platform and an almost identical antigen, a prefusion-stabilized SARS-CoV-2 spike protein-encoding mRNA. The vaccines differ in their proprietary lipid delivery platform and the amount of antigen used (30 μg for Pfizer and 100 μg for Moderna). Both vaccines provide exceptionally high protection, preventing about 95% of symptomatic COVID-19 disease and 100% of severe COVID-19 disease [4,5]. However, the induced immune responses are not long-lasting, and a booster is required after six months.

Single-dose vaccines have a significant advantage, as they allow for greater vaccination coverage and better patient compliance. The Johnson & Johnson (J&J, USA) Ad26.COV2.S vaccine is the only single-dose vaccine that has been evaluated and received FDA approval in the United States. The J&J vaccine comprises a recombinant, replication-incompetent human adenovirus type 26 (Ad26) vector encoding a full-length, membrane-bound prefusion-stabilized S protein. Although phase 3 clinical trial data reported a vaccine efficacy of 66.9% against moderate to severe-critical COVID-19, the vaccine immunity wanes after two months, requiring a booster dose [6,7] and limiting the utility of this vaccine candidate. Interestingly, a preclinical study with a single intranasal dose of a live-attenuated parainfluenza virus-vectored SARS-CoV-2 vaccine was protective in hamsters [8]. However, long-term responses are yet to be determined for this vector.

The high transmissibility of SARS-CoV-2 in the absence of sufficient vaccine coverage has led to the emergence of several variants. The World Health Organization (WHO) has recently classified two variants of concern, Delta and Omicron. Follow-up studies conducted during the emergence of the Delta variant have shown waning immunity for all approved vaccines, including those from Pfizer and Moderna, with the most extreme decline for the J&J vaccine [9]. Due to the prevalence of the Delta variant, the CDC recommends booster doses for all three approved vaccines. Scientists have also recommended differential dosing for infected versus naïve individuals, as a single dose of an mRNA vaccine in previously infected individuals provided antibody titers similar to two doses in a naïve person [10]. These results indicate the great success of vaccines in preventing COVID-19 but also suggest that new, cost-effective vaccines inducing long-lasting immunity are crucial.

Our previous studies demonstrated the efficacy of two doses of our inactivated COVID-19 vaccine, CORAVAX, in hamsters. CORAVAX is an adjuvanted inactivated vaccine generated using the SAD-B19 Rabies vaccine strain encoding the S1 subunit of SARS-CoV-2. This study evaluates the efficacy of a single dose of CORAVAX in a hamster model of severe disease. A single dose of adjuvanted CORAVAX induced high S1 and RBD IgG responses along with high virus-neutralizing antibodies against SARS-CoV-2. The single-dose vaccination protected vaccinated animals from weight loss and lung inflammation, reduced lung viral loads, and alleviated the lung cytokine storm.

## 2. Materials and Methods

### 2.1. Ethics Statement

The studies were carried out according to the recommendations described in the Guide for the Care and Use of Laboratory Animals of the National Research Council. Thomas Jefferson University (TJU) is an AAALAC-accredited institution, and the IACUC Committee approved all animal work of TJU (protocol number 02173). All efforts were made to minimize animal suffering, and all procedures involving potential pain were performed with the appropriate anesthetic or analgesic. The number of hamsters used was scientifically justified based on statistical analyses of virological and immunological outcomes. Six-to-eight-week-old golden Syrian female hamsters (Envigo) were anesthetized with 5% isoflurane before immunization, blood collection, and the SARS-CoV-2 challenge. Animals were monitored daily for weight loss and signs of disease. Five of the animals in each group were euthanized by an overdose of CO_2_ inhalation on day 3 (day 42), day 7 (day 46), and day 14 (53) p.c. for viral load determination and analysis.

### 2.2. Viruses

The SARS-CoV-2 strain used in this study is the first U.S. isolate SARS-CoV-2 USA_WA1/2020 from the Washington State patient identified on 22 January 2020. This virus was deposited by the Centers for Disease Control and Prevention and obtained through BEI Resources (Manassas, VA, USA), NIAID, NIH (SARS-Related Coronavirus 2, Isolate USA-WA1/2020, NR-52281). Virus stocks were obtained at passage 4 (GenBank: MN985325.1) and propagated in Vero E6 cells. The stock used in this study is passage 5.

### 2.3. Cells

Vero (ATCC, CCL81), Vero E6 (ATCC, CRL-1586), BSR (a clone of baby hamster kidney cells), HEK-293T (ATCC, CRL 3216), and effector cells (murine FcγRIV ADCC Bioassay Effector Cells, Promega-M1201) were used.

### 2.4. Vaccine Production and Purification

Recombinant RABV virions were recovered, purified, inactivated, and titered as described previously [11]. Briefly, X-tremeGENE 9 transfection reagent (Millipore Sigma, St. Louis, MO, USA; Cat# 6365809001) was used to co-transfect the full-length viral cDNA clone encoding CORAVAX and plasmids encoding RABV N, P, and L proteins and T7 RNA polymerase into BSR cells in 6-well plates (RABV). At eight days post-transfection, the supernatant was collected, filtered through a 0.22 μm membrane filter (Millipore, St. Louis, MO, USA), and titrated on Vero cells. The presence of recombinant virus was verified by immunostaining with a monoclonal antibody against the RABV nucleoprotein (FujiRebio, Malvern, PA, USA; Cat# 800–092) and polyclonal antiserum against the SARS-CoV/CoV-2 S1 domain (Thermo Fisher, Corning, NY, USA; Cat# PA581798). The filtered virus was then used to inoculate Vero cells seeded in Cellstack Culture Chambers (Thermo Fisher Scientific, Corning, NY, USA) and propagated in VP-SFM medium (Thermo Fisher Scientific, Corning, NY, USA) over 18 days. Supernatant collected on day 10 post-infection was filtered through 0.45 μm PES membrane filters (Thermo Fisher Scientific, Corning, NY, USA) and layered onto 20% sucrose in DPBS. Virions were sedimented by ultracentrifugation in an SW32 rotor for 1.5 h at 25,000 rpm. Viral particles were resuspended in phosphate-buffered saline (PBS) and inactivated with 50 μL per mg of particles of a 1:100 dilution of β-propiolactone (BPL, Millipore Sigma, St. Louis, MO, USA; Cat# P5648) in cold water. The absence of infectious particles was verified by inoculating BSR cells with 10 μg of BPL-inactivated viruses over three passages.

### 2.5. Vaccination and SARS-CoV-2 Challenge

Thirty hamsters were housed individually in microisolator cages. Each animal received 10 µg of vaccine or control mixed with 50 μL of SEPIVAC (Seppic, Courbevoie, France) adjuvant (50:50, by volume) brought up to 100 µL with 1× DPBS. Hamsters were anesthetized using isoflurane for vaccination, bleeding, and challenge procedures. The hamsters were bled on days 0 (pre-dosing), 14, and 28. Hamsters were vaccinated once on day 0 with 50 µL of formulated vaccine or control injected intramuscularly into each thigh of the hamster (100 µL total dose/animal). On day 39, hamsters were weighed and then challenged intranasally with 100 µL of 1 × 10^4^ PFU of SARS-CoV-2 Washington strain dropwise into each nostril. Post-challenge, the hamsters were weighed daily and assessed for symptoms. On days 3, 7, and 14 post-challenge, five hamsters from each group were weighed and humanely sacrificed under steady CO_2_ conditions. Cardiac blood was collected, and cervical dislocation was performed. The nasal turbinates, lungs, and kidneys were harvested for determining qPCR viral RNA titers and infectious viral load. One lung lobe was placed in 4% formaldehyde and placed on ice. The spleen was harvested for RNA isolation in tubes prefilled with ceramic beads and 1000 µL of TRIzol tubes.

### 2.6. Recombinant Proteins for ELISA

*Purification of the HA-tagged S1 protein from the supernatant of transfected cells for ELISA:* Sub-confluent T175 flasks of 293T cells (human embryonic kidney cell line, ATCC) were transfected with a pDisplay vector encoding amino acids 16 to 682 of SARS-CoV-2 S (S1) fused to a C-terminal hemagglutinin (HA) peptide using X-tremeGENE 9 reagent (Millipore Sigma, St. Louis, MO, USA; Cat# 6365809001). The supernatant was collected six days post-transfection, filtered through 0.22 um PES membrane filters (Thermo Fisher, Corning, NY, USA), and then loaded onto anti-HA agarose (Thermo Fisher Scientific, Corning, NY, USA, Cat# 26182) column equilibrated in PBS. After washing with ten bed volumes of PBS, the column was loaded with two column volumes of HA peptide at a 400 μg/mL concentration in PBS and incubated overnight at 4 °C. The following day, the protein was eluted with two column volumes of HA peptide followed by two column volumes of PBS. Fractions were collected and analyzed by Western blotting with polyclonal antiserum against the S1 domain (Thermo Fisher, Corning, NY, USA; Cat# PA581798). Peak fractions were then pooled and dialyzed against PBS in 10,000 molecular weight cutoff (MWCO) dialysis cassettes (Thermo Fisher Scientific, Corning, NY, USA; Cat# 66380) to remove excess HA peptides. After dialysis, the protein was quantitated by UV spectrophotometry and frozen in small aliquots at −80 °C.

*Purification of the RBD-His protein for ELISA*: The SARS-CoV-2 RBD-His-tagged plasmid was purchased from Bei Resources (Manassas, VA, USA; NR-52309). Sub-confluent T175 flasks of 293T cells (human embryonic kidney cell line) were transfected with the RBD-His-tagged plasmid using X-tremeGENE 9 reagent (Millipore Sigma, St. Louis, MO, USA; Cat# 6365809001). The supernatant was collected six days post-transfection and filtered through 0.22 μm PES membrane filters (Thermo Fisher Scientific, Corning, NY, USA). The 5 mL HisTALON cartridge (Clontech Laboratories, San Jose, CA, USA; Cat# 635683) column was equilibrated with ten column volumes of Equilibration Buffer (HisTALON Buffer Set, Clontech Laboratories, San Jose, CA, USA; Cat# 635651). The filtered supernatant was loaded onto the HisTALON cartridge (Clontech Laboratories, San Jose, CA, USA; Cat# 635683) column at a speed of 1 mL/min. After washing with 10 column volumes of Wash Buffer (prepared by mixing 6.6 parts of Elution Buffer with 93.4 parts of Equilibration Buffer of the HisTALON Buffer Set), the sample was eluted (at a flow rate of ~1 mL/min) with approximately 8 column volumes of Elution Buffer, collecting 1 mL fractions. The sample protein concentration was assessed by measuring the absorbance of the eluted fractions at 280 nm (Nanodrop, Thermo Fisher Scientific, Corning, NY, USA). Eluted fractions were analyzed by Western blotting with a mouse monoclonal RBD-specific antibody (InvivoGen, San Diego, CA, USA; Cat# srbdmab10). Peak fractions were then pooled and dialyzed against PBS in 10,000 molecular weight cutoff (MWCO) dialysis cassettes (Thermo Fisher Scientific, Corning, NY, USA). After dialysis, the protein was quantitated by UV spectrophotometry and frozen in small aliquots at −80 °C.

### 2.7. Enzyme-Linked Immunosorbent Assay

To determine antibody responses to the S protein of SARS-CoV-2, an indirect ELISA was developed utilizing purified S1 or receptor-binding domain (RBD) protein. The production of the recombinant proteins is described above. An indirect ELISA measured humoral responses to SARS-CoV-2 S1 and RBD proteins. We tested individual hamster sera by enzyme-linked immunosorbent assay (ELISA) for the presence of IgG specific to SARS-CoV-2 S1 or RBD. To test for anti-SARS-CoV-2 S1 humoral responses, we produced soluble S1 or RBD as described above. The two recombinant proteins were resuspended in coating buffer (50 mM Na_2_CO_3_ [pH 9.6]) at a concentration of 0.5 μg/mL of S1 or 1 μg/mL of RBD, and then they were plated in 96-well ELISA MaxiSorp plates (Thermo Fisher Scientific, Corning, NY, USA) at 100 μL in each well. After overnight incubation at 4 °C, plates were washed three times with 1× PBST (0.05% Tween 20 in 1× PBS), followed by the addition of 250 μL of blocking buffer (5% dry milk powder in 1× PBST) and incubation at room temperature for 1.5 h. The plates were then washed three times with PBST and incubated overnight at 4 °C with serial dilutions of sera (three-fold dilutions in triplicate) in 1× PBST containing 0.5% BSA. Plates were washed three times the next day, followed by the addition of HRP-conjugated goat anti-Syrian hamster IgG secondary antibody (Jackson Immunoresearch, West Grove, PA, USA; Cat# 107-035-142, 1:8000 in PBST) or mouse anti-hamster-IgG2/3-HRP (Southern Biotech, Birmingham, AL, USA; Cat# 1935–05, 1:8000 in PBST) or mouse anti-hamster-IgG1-HRP (Southern Biotech, Cat# 1940–05, 1:8000 in PBST) for 2 h at RT. After the incubation, plates were washed three times with PBST, and 100 μL of o-phenylenediamine dihydrochloride (OPD) substrate (Thermo Scientific, Corning, NY, USA; Cat#34006 and 34062) was added to each well. The reaction was stopped by adding 50 μL of 3 M H_2_SO_4_ per well. Optical density was determined at 490 nm (OD490) and 630 nm (OD630) using an ELX800 plate reader (Biotek Instruments, Inc., Winooski, VT, USA). Plates were incubated for 15 min (IgG) or 20 min (IgG2/3 or IgG1) with OPD substrate before stopping the reaction with 3 M H_2_SO_4_. Data were analyzed with GraphPad Prism (Version 8.0 g) using 4-parameter nonlinear regression to determine the titer at which the curves reach 50% of the top plateau value (50% effective concentration [EC_50_]).

### 2.8. SARS-CoV-2 Neutralizing Antibody Response

Sera collected from animals were tested for neutralizing capabilities against SARS-CoV-2. Briefly, serum samples were heat-inactivated (30 min at 56 °C) and then diluted at 1:10. The diluted sera were serially diluted two-fold in a 96-well plate, and 30 μL of each serum dilution was mixed with 30 μL of SARS-CoV-2-WA (100 PFU). The serum/virus mixtures were incubated for 1 h at room temperature; 60 μL of the serum/virus mixtures were then transferred to Vero E6 cell monolayers in flat-bottom 96-well plates and incubated for two days at 37 °C. Plates were fixed with 4% PFA and removed from containment. Plates were blocked and then stained with DAPI at 0.5 µg/mL and the SARS-CoV-2-specific human antibody CR3022 (produced in-house from plasmids: BEI, Manassas, VA, USA; NR-53260) conjugated with a Dylight 650 fluorophore at a concentration of 4 µg/mL in 1× DPBS with 5% FBS. Plates were imaged and read on a Cytation5 plate reader (Biotek Instruments, Inc., Winooski, VT, USA). The convalescent sera were collected from human volunteers 2–4 weeks post–Washington strain infection in early 2020.

### 2.9. Rabies Virus Neutralization by RFFIT

Rapid fluorescent focus inhibition tests (RFFITs) were performed as previously described [12]. In brief, mouse neuroblastoma cells (NA) cultured in RPMI media (with 5% FBS) were seeded in 96-well plates and incubated for 48 h. Individual hamster sera from days 0 and 28 were serially diluted 3-fold, starting at a 1:20 dilution. A prediluted mixture of Rabies virus (strain CVS-11, previously determined to achieve 90% infection in confluent NA cells) was added to each serum dilution. The US standard Rabies immune globulin (World Health Organization standard (WHO STD)) was used at 2 IU/mL starting dilution. The sera/virus or WHO STD/virus mixture was incubated for 1 h at 34 °C. The medium was then removed from the NA cell plate, replaced with the sera/virus mix, and incubated for 2 h at 34 °C. The sera/virus mixture was aspirated and replaced with a fresh medium post-infection. The plates were then incubated for 22 h (24 hours’ total infection) at 34 °C. After incubation, cells were fixed with 80% acetone, dried, and stained with fluorescein isothiocyanate Anti-Rabies N Monoclonal Globulin (Fujirebio, Malvern, PA, USA) for >4 h. Wells were assessed for percentage infection using a fluorescent microscope. Fifty percent endpoint titers (EPTs) were calculated using the Reed–Muench method and converted to international units per milliliter by comparing the sample 50% EPTs to that of the US standard Rabies immune globulin.

### 2.10. Tissue Processing

Animals were euthanized on days 3, 7, or 14 post-challenge (p.c.), and lungs and nasal turbinates were harvested. One lobe of the right lung, the right kidney, and the right nasal turbinates were placed in prefilled tubes containing 1 mL of viral transport medium for viral load analysis. The right lung (one lobe), left nasal turbinates, left kidney, and spleen were placed in prefilled tubes containing 1mL of TRIzol reagent. The left lung lobe was perfused with 4% Formaldehyde, placed in a conical tube filled with 6 mL of 4% Formaldehyde, and put on ice. The lung samples were placed at 4 °C for three days before processing for histology and staining.

### 2.11. Viral Load Determination

Tubes containing organs in viral transport medium were homogenized using the BeadBlaster 24R homogenizer, and tissue homogenates were aliquoted and stored at −80 °C. Tissue homogenates were titrated by TCID_50_ assay on Vero E6 cell monolayers (ATCC, Manassas, VA, USA) in 96-well plates to determine viral loads. Plates were incubated for 2 days at 37 °C before being fixed with 4% PFA and removed from containment. Plates were blocked and then stained with DAPI at 0.5 µg/mL and the SARS-CoV-2-specific human antibody CR3022 (produced in-house from plasmids: BEI, Manassas, VA, USA; NR-53260) conjugated with a Dylight 650 fluorophore at a concentration of 4 µg/mL in 1× DPBS with 5% FBS. Plates were imaged and read on a Cytation5 plate reader (Biotek Instruments, Inc., Winooski, VT). The limit of detection was determined by mixing 1000 PFU/mL, 100 PFU/mL, 50 PFU/mL, and 10 PFU/mL of a known titer SARS-CoV-2 WA strain virus with lung or tissue homogenates followed by TCID_50_ assay to calculate the limit of detection.

### 2.12. Viral RNA Copies by qRT-PCR

Tubes containing organs in TRIzol were homogenized in Bead Ruptor 12 (Omni International, Kennesaw, GA, USA). Then, 100 μL of tissue homogenate was mixed with 900 uL of TRIzol Reagent (Life Technologies, Carlsbad, CA, USA). The RNA extraction protocol for biological fluids using TRIzol Reagent was followed until the phase separation. The remaining RNA extraction was performed using the PureLink RNA Mini Kit (Life Technologies, Carlsbad, CA, USA). The quantity and quality (260/280 ratios) of RNA extracted were measured using NanoDrop (Thermo Fisher, Corning, NY, USA). SARS-CoV-2 envelope gene subgenomic mRNA cDNA was generated from RNA (BEI Resources, Manassas, VA, USA; NR-52285) by One-Step RT PCR (SuperScript IV, Thermo Fisher, Corning, NY, USA) with primers SARS-CoV-2 E sgmRNA IVT Fwd (5′- GAATTCTAATACGACTCACTATAGGGGATTAAAGGTTTATACCTTCCC-3′) and SARS-CoV-2 E sgmRNA IVT Rev (5′-GCTAGCTTAGACCAGAAGATCAGGAACTC-3′). The SARS-CoV-2 E sgmRNA standards were generated by in vitro transcription of the generated SARS-CoV-2 E sgmRNA cDNA using the MegaScript T7 Transcription kit (Thermo Fisher, Corning, NY, USA), followed by the MEGAclear Transcription Clean-Up Kit. Aliquots of 2 × 10^10^ copies/μL were frozen at −20 °C. Five microliters of RNA per sample was run in triplicate using the primers: SARS-CoV-2 E sgmRNA Fwd (5′-CGATCTCTTGTAGATCTGTTCTC-3′), SARS-CoV-2 E sgmRNA Rev (5′-ATATTGCAGCAGTACGCACACA-3′), and probe-FAM (5′-ACACTAGCCATCCTTACTGCGCTTCG-3′).

### 2.13. Cytokine qPCR

Duplicates of each lung RNA sample were assessed for the presence of several cytokines in a multiplex format. Three 5-Plex assays were generated: (1) γ-actin, IFN-γ, CCL17, CCL3, and CCL5, (2) γ-actin, CXCL10, IFN-α, IL10, and IL12 p40, and (3) γ-actin, IL17, IL21, and IL4. One 4-Plex assay was generated to assess γ-actin, IL-6, TGF-β1, and TNF-α. The mRNA assay primer and probe sequences are listed in Appendix A. The fold gene expression was calculated by 2^-ΔΔCt^ by calculating the ΔCt for vaccinated, control, and untreated hamster lungs. Of note, CXCL10 was not detectable in any animal.

### 2.14. Histology

Hamster lungs were harvested in 4% PFA and stored at 4 °C. The tissue was dehydrated, fixed, and infiltrated with paraffin on the Leica Polaris 2 tissue processor. The infiltrated tissue was then embedded and sectioned at 5 microns. Antigen retrieval was performed on the Roche Ventana Discovery ULTRA staining platform using Discovery CC1 (Roche, Indianapolis, IN, USA; Cat#950-500) for a total application time of 64 min. The primary antibody, human Anti-SARS-CoV-2 Nucleocapsid monoclonal (E16C) (Thermo Fisher, Corning, NY, USA; MA1-7403, 0.1 mg/mL), was incubated at a 1:100 dilution, at room temperature, for 45 min. Secondary immunostaining used a horseradish peroxidase (HRP) multimer cocktail (Roche, Indianapolis, IN, USA; Cat#760-500), and immune complexes were visualized using the ultraView Universal DAB (diaminobenzidine tetrahydrochloride) Detection Kit (Roche, Indianapolis, IN, USA; Cat#760-500). Slides were then washed with a Tris-based reaction buffer (Roche, Indianapolis, IN, USA; Cat#950-300) and counter-stained with Hematoxylin II (Roche, Indianapolis, IN, USA; Cat #790-2208) for 4 min. Scoring was performed as shown in Appendix A.

### 2.15. Antibody-Dependent Effector Functions

The assay was performed in 96-well, flat, white-bottom plates (Thermo Fisher, Corning, NY, USA) preseeded with 30,000 Vero CCL81 cells and infected after 24 h with Measles vaccine (Edmonston B strain) vector expressing full-length SARS-CoV-2 spike at an MOI of 0.5. After 60 h, the medium was removed, and 25 μL of assay buffer (RPMI 1640 with 4% low-IgG FBS) was added to each well. Sera were pooled from each group and heat-inactivated for analysis. Then, 25 μL sera at a starting dilution of 1:30 was serially diluted 3-fold in assay buffer (in duplicates). The sera were then incubated with the infected cells for 30 min at 37 °C. Post-challenge (day 15 post-challenge) hamster sera from the unvaccinated challenged group were used as a control. Genetically modified Jurkat cells expressing mouse FcγR IV with a luciferase reporter gene under the transcriptional control of nuclear-factor-activated T cell (NFAT) promoter were added at 1.5 × 10^5^ cells in 25 μL per well, which is approximately a 5:1 ratio of effector cells (Promega, Madison, WI, USA) to target cells. Cells were then incubated for another 6 h at 37 °C. After the incubation, 75 μL of Bio-Glo Luciferase assay reagent was added, and luminescence was quantified using a microplate reader. Fold induction was measured in relative light units and calculated by subtracting the background signal from wells without effector cells and dividing values for wells with antibodies by values for those with no antibody added. Specifically, fold induction was calculated: (RLU_induced_ − RLU_background_)/(RLU_uninduced_ − RLU_background_). The mean values and standard errors of the means (SEM) were reported, and a nonlinear regression curve was generated using GraphPad Prism 9.

### 2.16. Statistical Analysis

An unpaired two-tailed nonparametric t-test (Mann–Whitney) was performed on log-transformed data for each time point for the ELISA, virus-neutralizing antibody, viral load, and qPCR assays. An unpaired nonparametric two-tailed Mann–Whitney test was performed on percentage weights for each time point for the weight curves. *p* values are reported.

## 3. Results

### 3.1. Immunogenicity of a Single Dose of CORAVAX in Syrian Hamsters

We previously described the Rabies virus-vectored SARS-CoV-2 vaccine CORAVAX™ [11]. Utilizing the SAD-B19 Rabies vaccine strain, the S1 subunit of the SARS-CoV-2 spike protein (S) was incorporated into RABV virions. To promote the incorporation of the S1 domain, we constructed a gene encoding a fusion protein between 682aa of the N-terminus of SARS-CoV-2 S fused to a C-terminal domain of RABV glycoprotein (G) [11]. Two doses of inactivated CORAVAX were efficacious in hamsters and mice and induced high virus-neutralizing antibodies (VNA) [11,13].

Here, we evaluated the effectiveness of a single dose of CORAVAX in the Syrian hamster model. To this end, we vaccinated two groups of fifteen golden Syrian hamsters intramuscularly (i.m.) with either 10 μg of inactivated CORAVAX or the control vaccine FILORAB1 (Rabies-vectored Ebola vaccine) [14] adjuvanted with a squalene-based adjuvant (SEPIVAC SWE, Seppic). The animals were inoculated on day 0 and bled on days 0, 14, and 28 (Figure 1). On day 39, vaccinated and control hamsters were challenged intranasally with a dose of 10^4^ PFU of the SARS-CoV-2 isolate USA_WA1/2020. Five animals from each group were necropsied on days 3, 7, and 14 post-challenge.

To analyze the immunogenicity of the CORAVAX vaccine, the sera from both groups of hamsters were assayed for SARS-CoV-2-specific antibody responses by ELISA specific for SARS-CoV-2 S1 as well as the receptor-binding site (RBD) (Figure 2). We observed high SARS-CoV-2 S1 and RBD-specific IgG responses in the CORAVAX group as early as day 14 post-vaccination. These responses were significantly increased until day 28 (Figure 2A,B), indicating the long-term in vivo stability of the antigen. As previously reported for CORAVAX with another adjuvant, we observed a strong Th1-biased immune response on day 28, indicated by the induction of S1 IgG2/3 responses (Figure 2C). As expected, no SARS-CoV-2 S1 or RBD-specific immune response was detected before the challenge in the control animals vaccinated with the EBOV vaccine FILORAB1 (Figure 2A–C). These results suggest that a single inoculation may provide some level of protection.

We analyzed the sera post-challenge (p.c.). Of note, on day 3 p.c. (day 42), we observed S1 IgG and IgG2/3 and RBD IgG responses only in the CORAVAX-vaccinated animals, except one animal in the control group, which had a low level of RBD IgG responses (Figure 2A–C). On days 7 and 14 p.c. (days 46 and 53), we observed significantly increased S1 IgG responses in the CORAVAX group, which were approximately 64-fold (day 7) and 9-fold (day 14) higher than the responses seen in the control group (Figure 2A). Moreover, the CORAVAX-vaccinated animals induced a strong Th1-biased immune response at time points post-challenge, a feature previously shown to be beneficial in combatting SARS-CoV-2 infection [15]. In contrast, control animals immunized with FILORAB1 induced either no S1 IgG2/3 responses or low levels in the case of two animals only on day 14 p.c. (day 53) (Figure 2C). As previously reported, we detected no significant differences in immune responses against the RBD IgG responses on days 7 and 14 p.c. for all groups (Figure 2B).

While the ELISA data are helpful for following immune responses over time and analyzing the quality of such responses, we tested the potency of the sera to neutralize SARS-CoV-2 using a virus neutralization antibody (VNA) assay. Figure 3 shows VNA titers against SARS-CoV-2.

The CORAVAX group followed the trend seen in the S1 IgG ELISA, wherein high VNA titers were detected on day 28, maintained on day 3 p.c., and increased on days 7 and 14 p.c. We did not detect 100% VNA titers on day 14 p.c., which may reflect levels lower than our limit of detection (dilution 1:20) (Figure 3). However, VNA was present for CORAVAX-immunized animals before the challenge (day 28), and no VNA titers were observed in the FILORAB1 group. The FILORAB1 group had no detectable VNA titers on day 3 p.c. (similar to S1 IgG responses), but some VNAs were detected on days 3 and 7 after the challenge. Of note, VNA titers of the FILORAB1 group were significantly lower than those of the CORAVAX group. On day 14 p.c., CORAVAX-immunized animals showed a VNA response surpassing responses detected for human convalescent sera. No background neutralization was observed in the negative control human sera (Figure 3).

### 3.2. CORAVAX Induces Potent Immune Responses against RABV

In addition to the VNA detected for SARS-CoV-2, we also analyzed the VNA induced by the two vaccines against RABV and observed no significant differences between the CORAVAX and FILORAB1 groups (Appendix A). The CORAVAX and FILORAB1 vaccines induced strong RABV VNA titers exceeding the protective levels against RABV, i.e., 0.5 IU/mL.

### 3.3. A Single Dose of CORAVAX Protects the Hamsters from Weight Loss and Viral Burden in the Lungs Post-SARS-CoV-2 Challenge

As shown above, CORAVAX is immunogenic; therefore, the hamsters were challenged intranasally on day 39 with 10^4^ PFU of SARS-CoV-2 isolate USA_WA1/2020 and were monitored for up to 14 days. The CORAVAX-vaccinated animals showed significantly less weight loss than the FILORAB1 controls, which lost more than 6% weight (Figure 4). On days 3, 7, and 14 p.c., five hamsters in each study group were humanely euthanized, and lungs, nasal turbinates, spleen, and kidneys were harvested. The viral loads were determined by assaying RNA for viral copies of replicating virus for subgenomic (sg) mRNA by RT-qPCR assay (Figure 5A,B) and infectious viral loads by TCID_50_ assay (Figure 5C,D). Significantly lower sgmRNA and infectious viral loads were detected in the lungs of the CORAVAX-vaccinated animals on day 3 post-challenge (Figure 5A,C). On day 7 post-challenge, significantly lower sgmRNA was detected in the lungs of the CORAVAX group, but no detectable infectious virus was seen in either the vaccine or control groups (Figure 5A,C). On day 14 post-challenge, there was no detectable infectious SARS-CoV-2 in either group, although similarly low levels of sgmRNA were detected in both groups (Figure 5A,C). In the nasal turbinates, we only detected infectious virus on day 3 p.c., with no significant differences between the two groups (Figure 5B,D).

### 3.4. A Single Dose of CORAVAX Can Significantly Reduce Lung Pathology in Hamsters

Lung sections were collected from control and vaccinated animals on days 3, 7, and 14 p.c. and processed for histology (Figure 6, Figure 7 and Figure 8). The images and description of the histological changes identified in this study are provided in Appendix A. Sections were scored in a blinded manner (scoring criteria, Appendix A). On day 3 p.c., the control FILORAB1 specimens showed severe, patchy inflammation involving more than 50% of the examined lung with severe involvement of the airways (bronchitis/bronchiolitis), extending into the alveolar septa (pneumonitis) with alveolar edema and hemorrhage (Figure 6, Figure 7 and Figure 8). We also noted severe vasculitis. The immunohistochemical stain for the SARS-CoV-2 nucleoprotein antigen showed extensive multifocal positive cells involving bronchial epithelium, alveolar epithelium, and macrophages. In contrast, on day 3 p.c., the CORAVAX-vaccinated specimens showed only mild to moderate patchy inflammation involving less than 50% of the examined lung with mild to moderate airway inflammation, mild alveolar wall infiltration, congestion, and interstitial edema. We also noted focal mild vasculitis. The immunohistochemical stain for the SARS-CoV-2 nucleoprotein antigen showed only multifocal positive bronchial cells.

On day 7 p.c., the control FILORAB1 specimens still showed significant inflammation involving around 50% of the examined lung (Figure 6, Figure 7 and Figure 8). The disease is multifocal, nodular, and bronchiolocentric with moderate to severe involvement of the airways and alveolar wall infiltration, edema, congestion, and focal hemorrhage. We also noted mild to moderate vasculitis. The immunohistochemical stain for the SARS-CoV-2 nucleoprotein antigen showed multifocal positive cells involving both bronchial epithelia and alveolar epithelia but to a much lesser extent than those observed on day 3 p.c. On day 7, the CORAVAX-vaccinated group showed significant improvement, with only mild airway inflammation, mild alveolar edema, and congestion involving less than 30% of the examined lung. No significant vasculitis was noted. The immunohistochemical stain for the SARS-CoV-2 nucleoprotein antigen showed no definite positive cells.

On day 14 p.c., the control FILORAB1 specimens showed improvement, with inflammation involving less than 30% of the lung with mild to moderate airway inflammation, mild to moderate alveolar congestion, edema, and mild vasculitis (Figure 6, Figure 7 and Figure 8). The immunohistochemical stain for the SARS-CoV-2 nucleoprotein antigen showed no definite positive cells. On day 14 p.c., the CORAVAX-vaccinated group showed no significant airway inflammation, only mild alveolar congestion and edema, and no significant vasculitis. The immunohistochemical stain for the SARS-CoV-2 nucleoprotein antigen showed no definite positive cells.

### 3.5. A Single Dose of CORAVAX Can Significantly Prevent SARS-CoV-2-Induced Cytokine Storm in the Lungs

Uncontrolled induction of inflammation is one of the hallmarks of pathology after infection with SARS-CoV-2 [16]. Therefore, the lungs of the vaccinated and control animals were assayed for cytokine (IFN-γ, IFN-α, IL-10, IL-6, IL-4, IL-2, IL-21, IL-17, TGF-β1, and TNF-α) and chemokine (CCL5 and CCL3) expression by qRT-PCR (primer and probe sequences, Appendix A), normalized to γ-actin and expressed as fold changes over the average expression of three uninfected, unchallenged hamsters. On days 3 and/or 7 post-challenge, we observed significantly higher levels of inflammatory cytokines IFN-γ, IFN-α, IL-10, and IL-6 in the control groups than were detected in CORAVAX-immunized animals (Figure 9). CCL5 and IL-4 were only detectable in the control groups, suggesting a more substantial Th2 bias in the control group, as seen in the isotype-specific ELISA presented above (Figure 2C). On day 7 p.c., there was a significant upregulation of CCL3, IL-2, IL-21, and TGF-b1 in the CORAVAX-vaccinated animals, which may have contributed to dampening the inflammation and allowing for viral clearance, as observed in the lung histology (Figure 6C,D and Figure 8C,D). We saw no significant difference in the IL-17 and IL-12 p40 expression between the two groups after the challenge (Figure 9 and Appendix A. Of note, CXCL10 was not detectable in any animal.

### 3.6. A Single Dose of CORAVAX Induces Antibodies with Antibody-Dependent Cellular Cytotoxicity Activity

While we previously found high VNA titers in animals immunized with CORAVAX, we wanted to test if CORAVAX also induced non-neutralizing antibody functions. To understand the potential mechanism of S1 antibody-mediated protection, we tested pre-challenge hamster sera (day 28) from vaccinated and controlled animals for antibody-dependent cellular cytotoxicity (ADCC) activity using a recombinant measles virus expressing SARS-CoV-2 S in Vero-CCL81 cells. These reporter cells showed ADCC activation in the ADCC Reporter Bioassay, where the Fc portion of the antibody binds to the FcγRIIIa (mouse) of the effector cells (Jurkat) and results in a quantifiable luminescence signal from the NFAT (nuclear factor of activated T-cells) pathway. We observed the highest ADCC activity in the CORAVAX serum. In contrast, we observed no ADCC activity in post-challenge hamster sera (15 days p.c.) or the control FILORAB1 groups (Figure 10), suggesting that CORAVAX induced antibodies with neutralizing and non-neutralizing functions.

## 4. Discussion

Several SARS-CoV-2 vaccines are now approved and available to the public. Even though the currently approved vaccines have been instrumental in controlling and limiting the SARS-CoV-2 pandemic, there is a need for better vaccines in terms of the durability of protective responses, limited variant cross-protection, high production costs, and low-temperature stability of some vaccines, which render them less suitable for countries with less developed infrastructure. Furthermore, none of the current vaccines are approved for children below five years. The immunity provided by prior infection and the current vaccines is high for a few months but not sufficient to tackle emerging variants. The emergence of the Delta and Omicron strains highlights the possibility of reinfection and loss of protection against newly emerging strains, which will require the continued development of modified or new vaccines with broader coverage. While studies have verified waning immunity after six months for the mRNA vaccines (Pfizer and Moderna) and two months after the J&J vaccine, it is still debated whether a prior infection can provide long-term protection [17,18]. Although protection from reinfection decreases with time in previously infected individuals, it is higher than in those who received two doses of the Pfizer vaccine [19]. Maintaining public health measures to curb transmission as the COVID-19 pandemic continues is critical for preventing morbidity and mortality. Modifying guidelines as per immunity status to improve vaccine coverage may be required. Studies have shown that a single dose of the vaccine in previously infected individuals is as protective as, if not better than, two doses of the mRNA or a single dose of the AstraZeneca vaccine [20,21,22].

Most of the current WHO-approved vaccines (mRNA-1273, BNT162b2, and Ad26.COV2.S) utilize the prefusion-stabilized spike protein as the antigen [6]. For the mRNA vaccines, although the antigen is similar, they differ in their mRNA content (100 μg for mRNA-1273 vs. 30 μg for BNT162b2), the dosing interval between prime and boost vaccines (4 weeks for mRNA-1273 vs. three weeks for BNT162b2), and the lipid composition of the nanoparticles used for packaging the mRNA content [2]. Initial studies demonstrated similar efficacy for both mRNA vaccines. Among the WHO-approved replication-incompetent adenoviral vaccines, AstraZeneca’s ChAdOx1 nCoV-19 vaccine (AZD1222) and Jansen’s Ad26.COV2, AZD1222 expresses the full-length spike, while Ad26.COV2.S expresses a prefusion-stabilized spike protein. Both vaccines are efficacious (~66–76%) but to a lower extent than the mRNA vaccines [23]. Large-scale human adenoviral COVID-19 vaccinations appear to be associated with rare cases of vaccine-induced immune thrombocytopenia (VITT) that may be due to the interaction of the viral vector with platelet factor 4 (PF4) [24,25,26,27,28,29,30]. Recently, WHO gave emergency use approval to an adjuvanted recombinant nanoparticle vaccine from NOVAVAX, NVX-CoV2373, with an efficacy of 89.7% in phase 3 trials [31,32,33].

The emergence of variants in the presence of waning vaccine immunity has raised concerns worldwide. A major current issue is that all vaccines express the ancestral SARS-CoV-2 spike, whereas the currently circulating variants, including Delta and Omicron, have significant mutations in the spike-encoding region, rendering the vaccine less effective and giving rise to mutants. Generating and testing new variant vaccines may be required to control the spread of SARS-CoV-2. Single-dose variant vaccines that can induce long-term immunity may help to improve vaccine coverage and reduce reinfection rates. Testing vaccines in previously infected animals may become critical as infection rates rise worldwide. In addition, developing standardized assays to determine the virus-neutralizing titer (VNT) thresholds essential for protection may improve vaccine quality [34].

Here, we report the immunogenicity, efficacy, reduced lung inflammation, and potential mechanism of our single-dose S1-based COVID-19 vaccine, CORAVAX, in hamsters. The antigen used is a chimeric protein that contains the S1 domain (1-682aa) fused to the C-terminus of RABV-G for tethering to the Rabies virus virion. We previously analyzed the correct folding of our chimeric S1 in CORAVAX by verifying its binding to the hACE2 protein [11]. An S1-based vaccine, when folded correctly, might serve as a good immunogen, as it ensures immune responses against the important neutralizing epitopes identified in human convalescent patients that recognize RBD, NTD S1, and the quaternary epitope that bridges the two RBDs [35].

Our previous study showed that two doses of CORAVAX adjuvanted with MPLA-Addavax, a TRL4 agonist, induced high levels of neutralizing antibodies, generated a strong Th1-biased immune response (indicated by high S1 IgG2/3 responses), and were efficacious in hamsters with viral clearance in the nasal turbinates [13]. In human patients with mild/recovered COVID-19, a Th1-biased immune response is beneficial for protection against disease [36]. The current study used SEPIVAC SWE (developed by VFI and Seppic) adjuvant, a squalene-in-water emulsion, similar to Addavax (Invivogen) tested in our previous study. We observed a significant increase in antibody and VNTs from day 14 to day 28 with a single dose of adjuvanted inactivated CORAVAX, suggesting antigen stability. Of note, CORAVAX induced high 100% VNT titers. As previously published, CORAVAX induced a strong Th1-biased immune response, which helped significantly reduce inflammation and viral loads in the lungs. The controls induced a stronger inflammatory cytokine and chemokine response in the lungs on day 3 and day 7 post-challenge than the vaccinated animals, in addition to the Th2 cytokines only being induced in the control group. Sera from CORAVAX-vaccinated hamsters demonstrate antibody-dependent cytotoxic activity, suggesting a dual function of the antibodies. Therefore, we can conclude that a single dose of CORAVAX is efficacious and prevents weight loss and the cytokine storm seen in the control lungs.

The histopathologic tissue examination on days 3 and 7 p.c. showed that the CORAVAX-vaccinated animals had significantly reduced inflammation, as indicated by the accumulation of inflammatory cells compared to the controls. CORAVAX-vaccinated animals always had significantly lower average pathology lung scores, indicating excellent protection from lung disease. A significantly higher amount of antigen was detected in the lungs of the controls than in the CORAVAX-vaccinated animals on days 3 and 7 p.c. By day 14 p.c., the inflammation was completely cleared in the vaccinated animals and significantly reduced in the control unvaccinated group, and the antigen was absent in the lungs of both groups, which is a common feature of the hamster model. The hamster model is well established for SARS-CoV-2, as it mimics the human lung pathology seen in COVID-19 patients [16]. This study concludes that SARS-CoV-2-challenged CORAVAX-vaccinated animals never developed the severe inflammation associated with the disease (compared with the control animals). We also observed a rapid decline and early elimination of the SARS-CoV-2 nucleoprotein antigen in the vaccinated animals (compared with the unvaccinated counterparts). These findings are a strong indication of protection of the lungs by the CORAVAX vaccine.

This study is the first to show the efficacy of a single-dose RABV-vectored COVID-19 vaccine, CORAVAX, in the Syrian hamster model of severe disease. The Rabies vaccine vector has significant advantages, including (1) an excellent safety profile, (2) approved administration to pregnant women, children, and the immunocompromised, (3) the induction of lifelong immunity, (4) amenable to boosting several times, and (5) less reactogenicity [37]. This study showed significant protection with a single dose of CORAVAX, yet it is evident that two doses are required for complete viral clearance from the nasal turbinates. Studies evaluating the longevity of immune responses after a single dose and the boosting potential of CORAVAX in previously infected hamsters are yet to be assessed. Production of CORAVAX is facile, as it would follow the existing RABV vaccine manufacturing facilities and technologies. To further evaluate CORAVAX, a phase I human clinical trial is being conducted in India by Bharat Biotech International Ltd., with plans for a follow-up phase I study in the US [38]. To summarize, CORAVAX is a promising COVID-19 vaccine candidate with an excellent safety profile and proven efficacy in several animal studies.

## Figures and Tables

**Figure 1 viruses-14-01126-f001:**
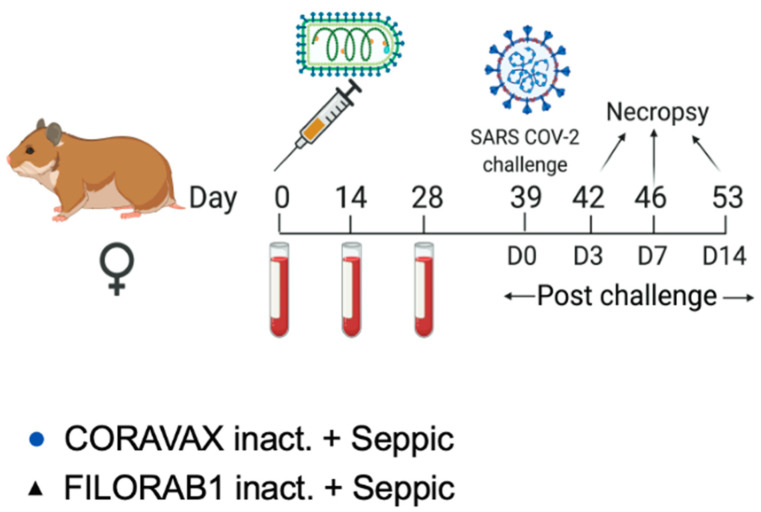
**Vaccination schedule**. Syrian hamsters were immunized on day 0 with 10 μg of chemically inactivated CORAVAX or FILORAB1 with Seppic (SEPIVAC SWE) adjuvant. The animals were challenged on day 39 with a dose of 10^4^ PFU of the SARS-CoV-2 isolate USA_WA1/2020. Serum was collected from each hamster on days 0, 14, and 28 and at necropsy (days 42, 46, or 53) for analysis. Animals were necropsied on days 42 (day 3 p.c.), 46 (day 7 p.c.), and 53 (day 14 p.c.). Created with BioRender.com.

**Figure 2 viruses-14-01126-f002:**
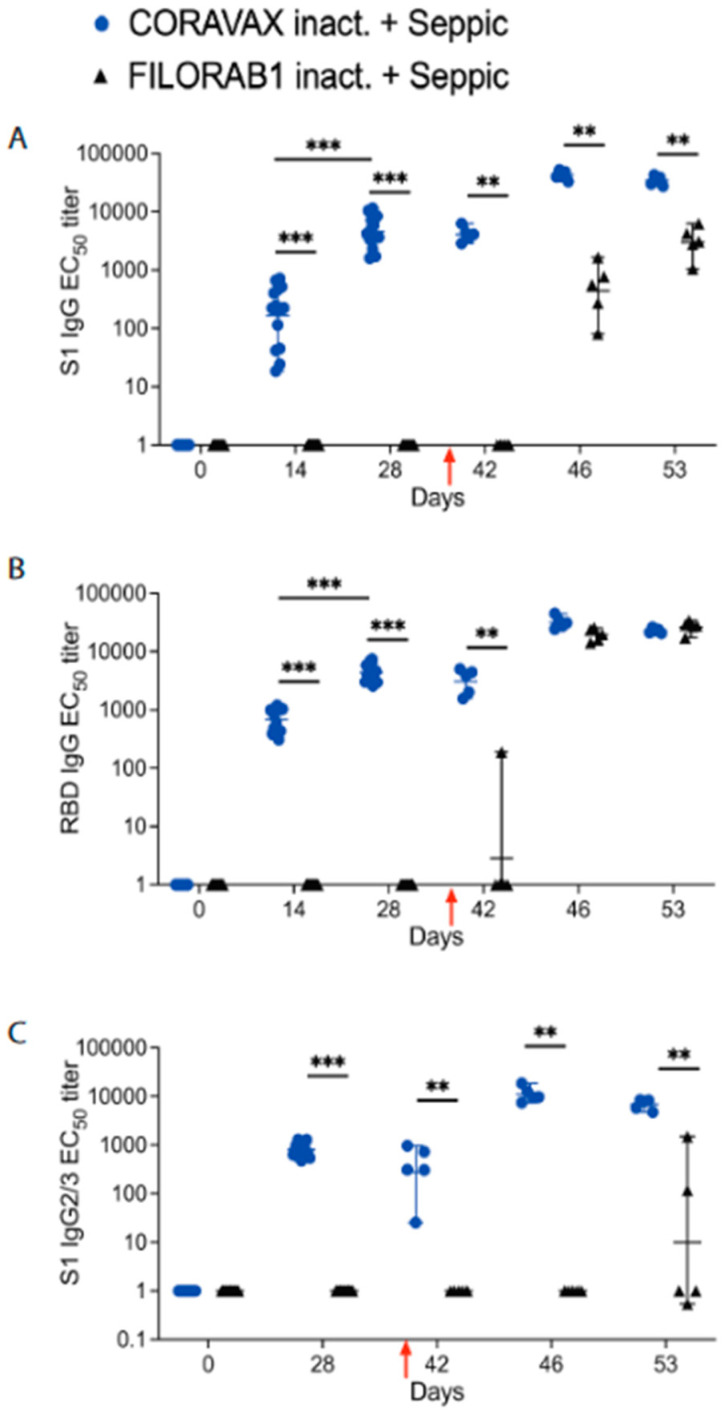
**SARS-CoV-2 immune responses.** Serum samples collected from each hamster were evaluated for SARS-CoV-2 S-specific immune responses by (**A**) ELISA, anti-SARS-CoV-2 S1 IgG responses represented as EC50 titers over time, (**B**) ELISA, anti-SARS-CoV-2 RBD responses, and (**C**), ELISA, anti-SARS-CoV-2 S1 IgG2/3 responses. The CORAVAX vaccine group is shown in blue circles, and the FILORAB1 group is in black triangles. For (**A**–**C**), mean titers ± SD are depicted for each group per time point, and *p* values were determined by Mann–Whitney test. Only significant differences are depicted. *p values are defined as: p* > 0.123 (ns), *p* < 0.033 (*), *p* < 0.002 (**), *p* < 0.001 (***). Red arrow indicates the day of challenge (day 39).

**Figure 3 viruses-14-01126-f003:**
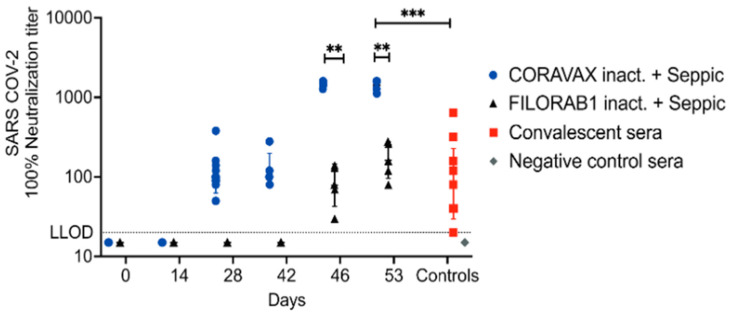
**SARS-CoV-2 virus-neutralizing antibody responses.** Serum samples collected from each hamster were evaluated for SARS-CoV-2 S-specific virus-neutralizing antibody (VNA) responses using 100 PFU of SARS-CoV-2 isolate USA_WA1/2020.100%. VNA titers are depicted for hamster sera at different time points along with convalescent sera (*n* = 7) and negative control sera (*n* = 7). The groups are depicted with the following symbols: CORAVAX group in blue circles, FILORAB1 control group in black triangles, convalescent sera in red squares, and negative control sera in grey diamonds. Mean titers ± SD are depicted for each group per time point, and *p* values were determined by Mann–Whitney test. Only significant differences are depicted. *p values are defined as: p* > 0.123 (ns), *p* < 0.033 (*), *p* < 0.002 (**), *p* < 0.001 (***). LLOD stands for lower limit of detection.

**Figure 4 viruses-14-01126-f004:**
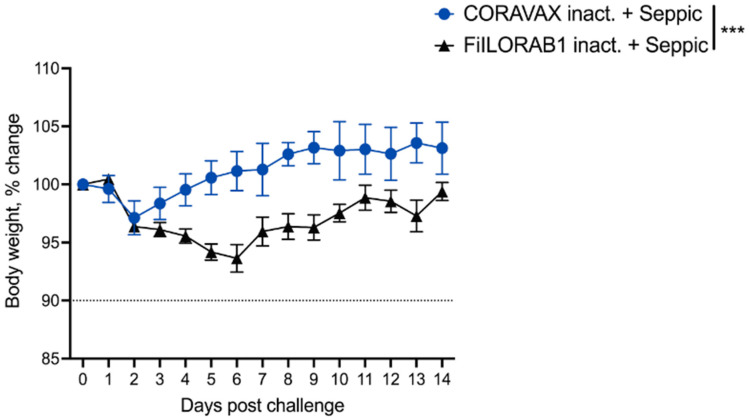
**Hamster body weight after SARS-CoV-2 infection**. Hamsters were vaccinated on day 0 and challenged intranasally with 10^4^ PFU SARS-CoV-2 on day 39. Percent change in body weight. CORAVAX vaccine group is shown in blue, and FILORAB1 group is in black. *n* = 15 for FILORAB1 group (5 hamsters each euthanized on days 3, 7, and 14 p.c.) and *n* = 15 for CORAVAX group (5 hamsters each euthanized on days 3, 7, and 14 p.c.). Body weight *p* value determined by Wilcoxon test; *p* > 0.123 (ns), *p* < 0.033 (*), *p* < 0.002 (**), *p* < 0.001 (***).

**Figure 5 viruses-14-01126-f005:**
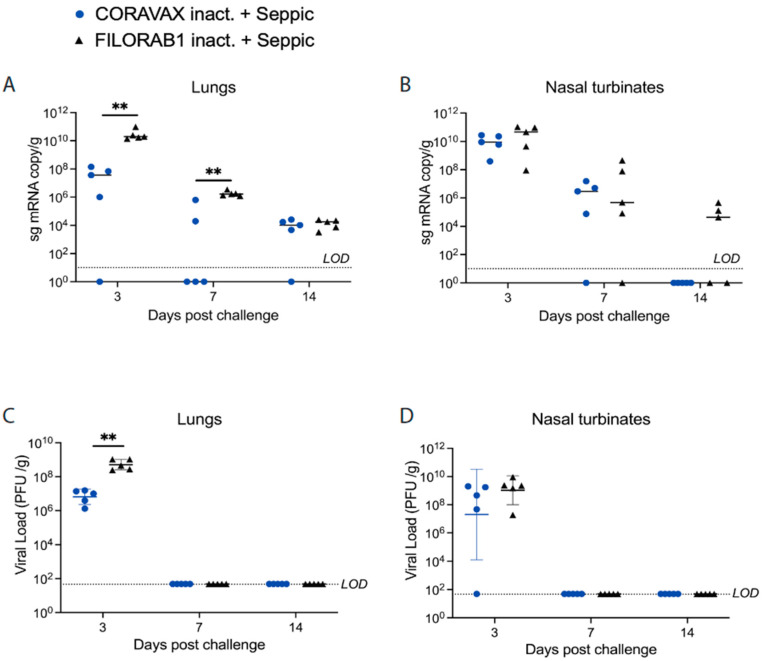
**SARS-CoV-2 tissue viral load in hamsters.** Hamsters were challenged intranasally with 10^4^ PFU SARS-CoV-2, and five of the animals in each group were euthanized on days 3, 7, and 14 p.c. Right lungs (**A**,**C**) and nasal turbinates (**B**,**D**) from each animal were homogenized in media, and viral loads were determined by qRT-PCR (A, B) or by TCID_50_ assays on Vero E6 cells (**C**,**D**). The limit of detection for the qRT-PCR assay is 10 copies. The limit of detection for the plaque assay is 50 PFU/mL per lung/nasal turbinate tissue. The CORAVAX vaccine group is shown in blue, and the FILORAB1 group is in black. Data represent mean ± S.D., *n* = 5 for FILORAB1 group days 3, 7, and 14 time points, and *n* = 5 for CORAVAX group days 3, 7, and 15 time points; *p* values were determined by Mann–Whitney test; *p* > 0.123 (ns), *p* < 0.033 (*), *p* < 0.002 (**), *p* < 0.001 (***). Only significant differences are depicted.

**Figure 6 viruses-14-01126-f006:**
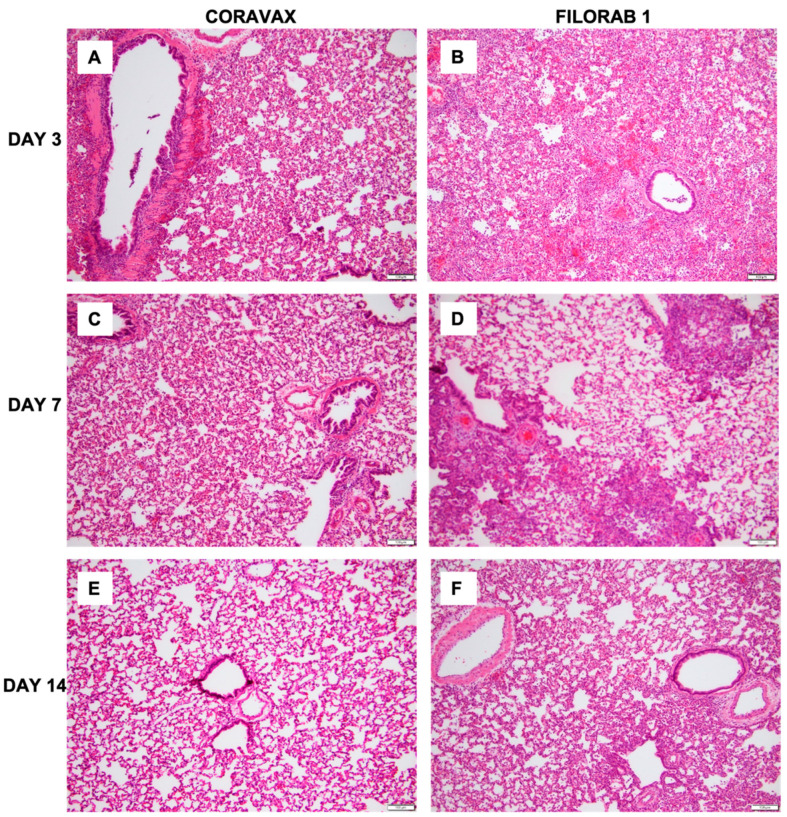
**SARS-CoV-2 lung pathology.** Representative histological images of SARS-CoV-2 infection in control and vaccinated hamster lungs. (**A**,**C**,**E**): CORAVAX vaccinated on days 3, 7, and 14 p.c. (**B**,**D**,**F**): FILORAB1 control on days 3, 7, and 14 p.c. All images were taken with 10× objective and 0.5 camera adaptor; the scale bar in every image equals 100 μm. The images show significantly less lung inflammation in the vaccinated animals than in the control unvaccinated animals. The difference is more prominent on day 3 p.c., followed by day 7 p.c.

**Figure 7 viruses-14-01126-f007:**
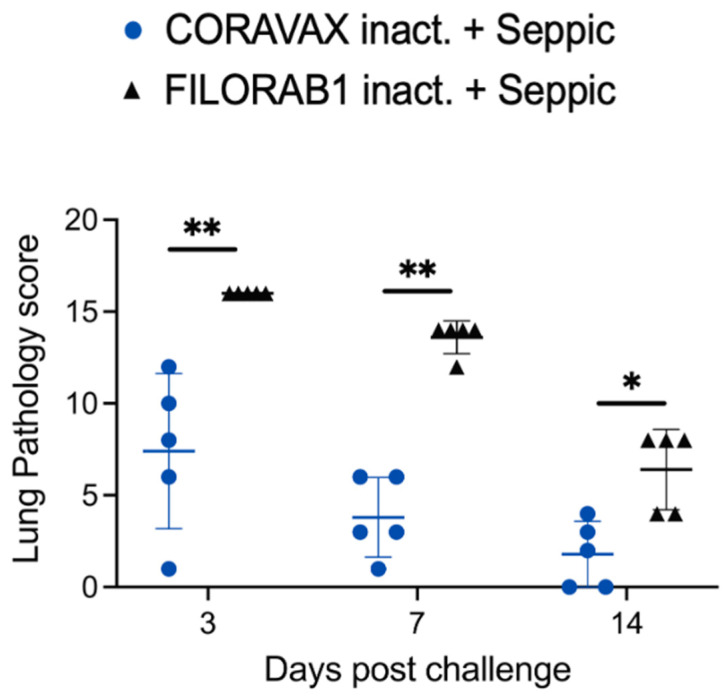
**Comparative pathology scores for lungs from CORAVAX-vaccinated and control hamsters post-SARS-CoV-2 challenge.** Overall lung pathology scores on days 3, 7, and 14 days p.c. The pathology scores (mean) were calculated based on the criteria described in Appendix A. The CORAVAX vaccine group is shown in blue circles, and the FILORAB1 control group is in black triangles. Data represent mean ± S.D., *n* = 5 for FILORAB1 group and CORAVAX group for days 3, 7, and 15 time points; *p* > 0.123 (ns), *p* < 0.033 (*), *p* < 0.002 (**), *p* < 0.001 (***). Only significant differences are depicted.

**Figure 8 viruses-14-01126-f008:**
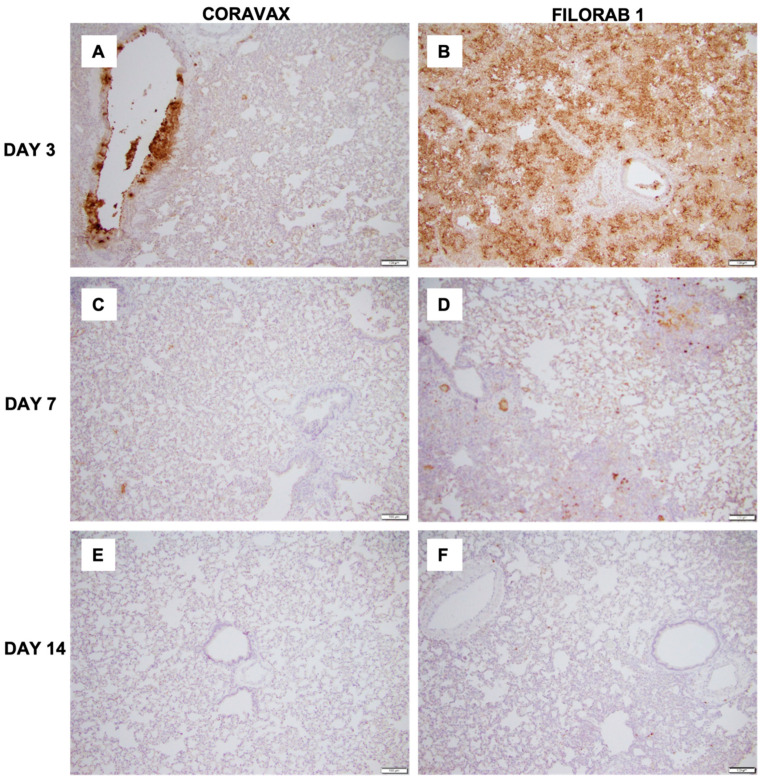
**SARS-CoV-2 nucleoprotein antigen lung staining.** Representative histological images of SARS-CoV-2 nucleoprotein (N) antigen staining in the lungs of control and vaccinated hamsters. (**A**,**C**,**E**) CORAVAX days 3, 7, and 14 p.c. (B,D,F) FILORAB1 days 3, 7, and 14 p.c. The presence of the SARS-CoV-2 N antigen is indicated by the brown DAB (3, 3′-diaminobenzidine) stain in the lungs. A significantly higher amount of N antigen staining is observed in the FILORAB1 lungs (**B**) than in CORAVAX lungs (**A**) on day 3 p.c. On day 7 p.c., immunohistochemical stain for the SARS-CoV-2 nucleoprotein antigen shows scattered positive cells in the control group (**D**) and no definite positive cells in the vaccinated group (**C**). SARS-CoV-2 N antigen staining is absent in the FILORAB1 (**F**) and CORAVAX lungs (**E**) on day 14 p.c. All images were taken with 10× objective and 0.5 camera adaptor; the scale bar in every image equals 100 μm.

**Figure 9 viruses-14-01126-f009:**
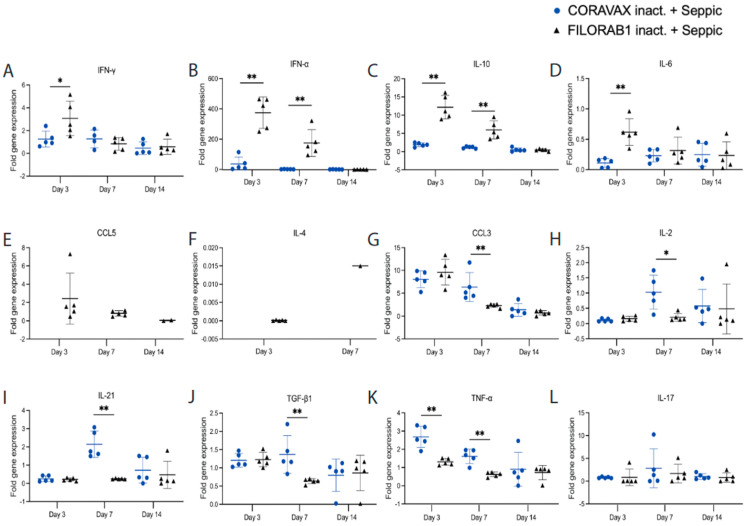
**A single dose of CORAVAX immunization prevents a cytokine storm in the lungs**. The CORAVAX vaccine group is shown in blue circles, and the FILORAB1 control group is in black triangles. Total RNA was extracted from the lungs of hamsters necropsied on days 3, 7, and 14 after challenge with SARS-CoV-2. Hamster (**A**) IFN-γ, (**B**) IFN-α, (**C**) IL-10, (**D**) IL-6, (**E**) CCL5, (**F**) IL-4, (**G**) CCL3, (**H**) IL-2, (**I**) IL-21, (**J**) TGF-β1, (**K**) TNF-α, and (**L**) IL-17. Cytokine/chemokine mRNAs were quantified by real-time RT-PCR. γ-Actin mRNA was used as an internal control. Data are shown as fold change in gene expression compared to normal animals (unimmunized and unchallenged) after normalization. Data were analyzed using Mann–Whitney test for each time point (* *p* < 0.05; ** *p* < 0.01; *** *p* < 0.001).

**Figure 10 viruses-14-01126-f010:**
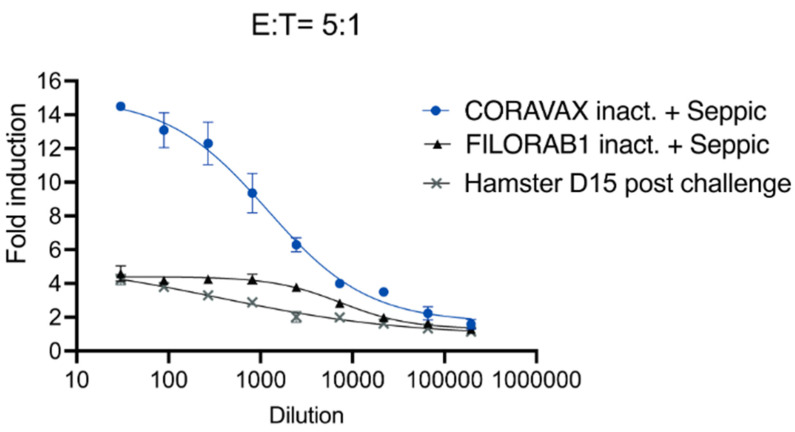
**A single dose of CORAVAX immunization induces antibody-dependent cellular cytotoxicity activity.** Antibody-dependent cellular cytotoxicity (ADCC) activation activity luciferase reporter assay. Vero cells infected with a Measles virus-expressing full SARS-CoV-2 spike protein were incubated with hamster sera and then incubated with effector cells (mouse FcγRIV-expressing Jurkat cells at a ratio of 5:1 (effector to target)). Heat-inactivated pooled sera were used for the assay. The CORAVAX vaccine group is shown in blue circles, the FILORAB1 control group is in black triangles and fifteen days post-challenge hamster sera from a previous experiment is depicted with a grey multiplication sign. The figure represents one replicate of 3 repeats. Fold induction is depicted for each group.

## Data Availability

The data presented in this study are available in the article or Appendix A presented here.

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

Hamsters. Cell Rep. Med..

