# Peer review of "A Single Dose of the Deactivated Rabies-Virus Vectored COVID-19 Vaccine, CORAVAX, Is Highly Efficacious and Alleviates Lung Inflammation in the Hamster Model"

_viruses, 2022, doi:10.3390/v14061126_

Round 1
Reviewer 1 Report
This manuscript by Kurup and colleagues evaluates the potency of a novel SARS CoV2 vaccine candidate, CORAVAX, an adjuvanted, single dose rabies vector-based alternative. The vaccine was tested in hamsters, and all measures tested indicate strong protection from disease in animals subsequently exposed to infection. The manuscript is clearly written, the studies are straightforward and compelling, and the results are a significant step forward in the development of SARS CoV2 vaccine alternatives. A small number of issues should be addressed:
1. Throughout the manuscript, the authors refer to “waning immunity” to the initial (mostly RNA-based) CoV2 vaccines. Is this accurate? Surely antibodies decline, but are there studies to suggest that this decline is associated with increased risk of serious disease? The likely persistence of memory B and T cells may mean that measuring antibody titers alone is insufficient to claim a diminished host response.
2. A sentence in the Intro beginning “Several countries began administering vaccines…” seems either irrelevant, or out of place.
3. There is an intriguing note in the Materials and Methods that “CXCL-10 was not detected in any of the animals.” Should this observation be in the Results section? Is it known if hamsters make a functional CXCL-10 protein?
4. There is no Figure 1B, although the legend refers to one.
5. The pathology slides could benefit from some “guidance” from the authors, using arrows/arrowheads on the images to point out salient features that distinguish the CORAVAX pathology from the FILORAB1 tissues.
6. Figure 8B/D: The resolution of virus (or, at least, viral antigen) in the lungs of the unvaccinated animals is remarkable. This need not be done for this paper, but I’m curious if the resolution was accompanied by substantial cell death as well.
7. On page 17, in the section “A single dose of CORAVAX…”, there appears to be some copy/paste editing problem. Also, on page 18, there is a typo “Sike” instead of “Spike”.
8. In the Discussion, the sentence “The immunity provided by prior infection is high but not complete” could be confusing.
Reviewer 2 Report
This article described the data obtained with CORAVAX when used as a single dose. This work is a follow up of the CORAVAX story which has been already published in 2020. The experiments are well designed, data are convincing and well presented. This work is of strong interest for the researchers in the vaccine field on SARS-CoV-2.
Minor points:
1) Legend Figure 2: please add what the red arrow means
2) Figure 6: To help the readers, please provide a small "cartoon" showing how severe patchy inflammation looks like, the location of bronchitis, bronchiolitis, alveolar septa, bronchial epithelium, alveolar epithelium and so on.
Reviewer 3 Report
The authors investigated the efficacy of inactivated recombinant RABV based CORAVAX vaccine in syrian hamster model. Overall the study is well conducted and the manuscript is well written. Few concerns however remain
1.Figure 3, whats the rationale for comparison of VNA titers with human convalescent sera? Human titers vary by the time of sera collection and the variant they convalesced from.
2. Figure 2, describing fold increase/decrease in titers during longitudinal sera sampling would be helpful in the text rather than just "significant increase" descriptions in the results section
3. Figure 6 and 8, I would recommend marking areas of inflammation, vasculitis congestion and edema with different signs (arrows, asterisks etc) in histochemistry panels
4. Did the authors test cross neutralization of sera against different variants? Omicron BA.1 or BA.2?
